# Long-Term Outcome Following Liver Transplantation for Primary Hepatic Tumors—A Single Centre Observational Study over 40 Years

**DOI:** 10.3390/children10020202

**Published:** 2023-01-22

**Authors:** Christoph Leiskau, Norman Junge, Frauke E. Mutschler, Tobias Laue, Johanna Ohlendorf, Nicolas Richter, Florian W. R. Vondran, Eva-Doreen Pfister, Ulrich Baumann

**Affiliations:** 1Pediatric Gastroenterology, Department of Pediatrics and Adolescent Medicine, University Medical Centre Göttingen, Georg August University Goettingen, 37075 Goettingen, Germany; 2Pediatric Gastroenterology, Hepatology and Liver Transplantation, Department of Pediatric Kidney, Liver and Metabolic Diseases, Hannover Medical School, 30625 Hannover, Germany; 3Department of General, Visceral and Transplant Surgery, Hannover Medical School, 30625 Hannover, Germany; 4Institute of Immunology and Immunotherapy, University of Birmingham, Birmingham B15 2TT, UK

**Keywords:** hepatoblastoma, liver malignancy, pediatric liver transplantation, salvage transplant, AFP, tumor recurrence, long-term outcome, biliary complication

## Abstract

The incidence of pediatric liver tumors in general has been rising over the last years and so is the number of children undergoing liver transplantation for this indication. To contribute to the ongoing improvement of pre- and post-transplant care, we aim to describe outcome and risk factors in our patient cohort. We have compared characteristics and outcome for patients transplanted for hepatoblastoma to other liver malignancies in our center between 1983 and 2022 and analysed influential factors on tumor recurrence and mortality using nominal logistic regression analysis. Of 39 children (16 f) who had transplants for liver malignancy, 31 were diagnosed with hepatoblastoma. The proportion of malignant tumors in the transplant cohort rose from 1.9% (1983–1992) to 9.1% in the current decade (*p* < 0.0001). Hepatoblastoma patients were transplanted at a younger age and were more likely to have tumor extent beyond the liver. Post-transplant bile flow impairment requiring intervention was significantly higher compared to our total cohort (48 vs. 24%, *p* > 0.0001). Hearing loss was a common side effect of ototoxic chemotherapy in hepatoblastoma patients (48%). The most common maintenance immunosuppression were mTor-inhibitors. Risk factors for tumor recurrence in patients with hepatoblastoma were higher AFP before transplant (AFP_pre-LTX_), a low ratio of AFP_max_ to AFP_pre-LTX_ and salvage transplantation. Liver malignancies represent a rising number of indications for liver transplantation in childhood. Primary tumor resection can spare a liver transplant with all its long-term complications, but in case of tumor recurrence, transplantation might have inferior outcome. The rate of acute biopsy-proven rejections and biliary complications in comparison to our total transplant cohort needs further investigations.

## 1. Introduction

Hepatoblastomas are the most common malignant liver tumors in pediatric patients. Even though surgical resection is possible in up to 70–85% [1,2], around 20% of hepatoblastoma tumors remain non-resectable after state-of-the-art chemotherapy, and liver transplantation is the only curative option.

The incidence of primary liver malignancies in general [3,4], and as indication for pediatric liver transplantation, has significantly increased over the last years and liver tumors currently represent around 10% of indications for transplantation in the large pediatric liver centers in Europe [5,6] and the U.S. in this decade [7,8]. The reasons for this, beside a higher overall incidence, may include an improvement of neoadjuvant oncologic therapy and earlier detection of patients who might need a primary liver transplant approach.

As a consequence of steady improvements in pediatric oncologic care, chemotherapy and surgical liver transplant techniques, outcome in patients transplanted for hepatoblastoma has significantly improved over the last years and by far exceeds the results for adult liver transplantation for liver tumors, with 5-year survival rates reaching 72–86% in recent studies [9,10,11,12]. The introduction of the pre-treatment extent of the tumor (PRETEXT) and post-treatment extent (POSTTEXT) classification, which is now used in most liver centers world-wide, has helped to improve the identification of patients who need a hemi-hepatectomy or are recommended for primary liver transplantation after chemotherapy [13].

However, long-term outcomes are still significantly worse when compared to pediatric liver transplantation in general, where 5-year survival exceeds 95% in contemporary cohorts. There is scarce data regarding long-term outcome and risk factors for mortality and tumor recurrence, which is the most common cause of death in these children [9,11].

The decision for and timing of liver transplantation compared to partial surgical liver resection, in the context of neo-adjuvant chemotherapy in many cases, is based on tumor extent, surgical resectability and predicted biological chemosensitivity of the tumor. Pre-emptive liver transplantation in possibly resectable tumors should be avoided, but on the other hand, a primary non-transplant approach leading to recurrence of the tumor might lead to worse results than upfront transplantation. There are inconsistent results in studies comparing the outcome of these salvage transplants to primary transplantation after chemotherapy. A review summarizing the literature up to 2016 showed a considerably lower survival for patients with salvage transplant (41%) compared to primary transplants (85%) [14], but some recent studies describe no difference in survival [11,15].

A recent large study describing outcome and risk factors for 175 pediatric patients who had transplants due to liver tumors in North America identified AFP decrease, low post-transplant AFP in hepatoblastoma patients and younger age in patients with HCC as factors favoring event-free survival [11]. Other risk factors described are a lower total AFP at the time of diagnosis [16], vascular invasion [7], response to chemotherapy [15] and transplant era [9].

Further, cisplatin-induced hearing loss is a common complication in childhood cancer survivors and is described in more than 40% of all patients receiving cisplatin and from 38 to 65% of patients who have been treated with cisplatin for hepatoblastoma, underlining the importance in this special group of patients who receive comparably high doses of cisplatin; ototoxicity is clearly associated with cisplatin cumulative dose [14,15].

Comparable current data from European centers are limited, and should be collected to account for differing treatment protocols and transplant listing criteria and procedures in the EUROTRANSPLANT region.

Our aim was to describe the long-term outcome of pediatric liver transplant recipients transplanted for liver tumors, and to identify risk factors for mortality and tumor recurrence to finally improve clinical decision-making in this special group of pediatric hepatologic and oncologic patients.

## 2. Patients and Methods

This is a single center retrospective study.

### 2.1. Inclusion/Exclusion Criteria

All pediatric (age < 18 years at the day of transplantation) liver transplantations (pLTx) at Hannover medical school, taking place between 1 January 1983 and 1 October 2022, were reviewed (n = 910). Some 45 patients who received liver transplantation for a liver tumor or with a liver tumor in the explanted liver were identified. Patients with another chronic liver disease (PFIC I, PFIC II, PFIC IV, Tyrosinemia, two patients with Alagille syndrome) who developed a liver tumor (HCC in all cases) based on their underlying disease were excluded to prevent a bias in long-term outcome, as the transplant indication was the chronic liver disease. None of the six patients received neo-adjuvant or adjuvant chemotherapy, and in four of the six patients, HCC was an incidental finding in the explanted liver. Further on, we excluded patients in whom a malignancy was an incidental finding at transplantation and therefore not the indication for transplantation. Altogether, 39 patients who were transplanted for primary malignant liver disease were identified.

### 2.2. Data Acquisition and Statistical Analysis

Data were retrieved with our patient data processing program SAP and from the pediatric liver transplant database. Data was stored anonymously in MS Excel files and SAS JMP. Data was analysed by JMP.

The development of incidence of pediatric liver transplantation for malignoma over time was analysed with Pearson’s chi-square-test and corrected for the hypothetic probabilities according to the total number of pediatric liver transplants in our center.

The continuous variables of the baseline characteristics for the group of hepatoblastoma and other malignoma were tested for significant differences by pooled t-test and HSU/MCB, the categorical variables were tested using Pearson’s chi2-test, the effect of ordinal values such as era of transplantation was tested with ordinal logistic regression.

The effect of categorical variables on survival and tumor recurrence were analysed by nominal logistic regression analysis, the effect of continuous variables was analysed using Cox regression proportional hazard analysis. Hazard ratios for continuous variables are stated as per change of one unit of the regressor.

The effect of number of cisplatin cycles on hearing loss was only calculated for the hepatoblastoma patients using ordinary logistic regression.

*p*-values below 0.05 were considered statistically significant.

### 2.3. Definitions

The time of transplantation was divided into four eras of the same time span (1983–1992; 1993–2002; 2003–2012; 2013–2022). Please note that the observation period in 2022 only lasted until 1 October.

Hepatoblastoma extent was staged according to the PRETEXT/POSTTEXT staging system, that has lately been revised in 2017 and includes four groups that are based on four liver sections (left lateral, left medial, right anterior and right posterior) that would have to be removed to remove the whole tumor [13,17]:

PRETEXT I: Involvement of only either right posterior or only left lateral section.

PRETEXT II: Involvement of right anterior section only or left medial section only or caudate lobe only or combination of two sections as in right anterior + right posterior, left medial + left lateral, right posterior + left lateral.

PRETEXT III: Involvement of three liver sections either multifocal or as one tumor

PRETEXT IV: Involvement of all four liver sections either multifocal, diffuse or as one tumor.

Tumor extension surpassing the liver was defined as a proven infiltration of other organs or lymph nodes at any time before treatment or at any time until liver transplantation.

Salvage transplantation was defined as liver transplantation because of recurrent malignant disease after previous treatment by surgical tumor removal.

Secondary malignoma was defined as a malignant disease occurring any time after liver transplantation. Recurrence of tumor was the recurrence of primary tumor or tumor metastases after pediatric liver transplantation.

Hearing loss was defined as the requirement of hearing aids after reliable hearing test. Renal impairment was defined as a glomerular filtration rate below 60 mL/min (as calculated using the formula of Schwartz).

Biliary complication was defined as biliary leak, bilioma or biliary stricture (anastomotic or nonasanstomotic) requiring intervention (ERCP, PTCD or surgical intervention).

AFP_max_ was defined as the highest measured AFP value during the course of disease before liver transplantation. AFP_prepLTx_ was defined as the last documented AFP value before liver transplantation, in most of the patients within 2 days before liver transplantation.

### 2.4. Ethical Considerations

All caregivers of the patients have been asked for permission of anonymous retro-spective analyses for scientific purposes with admission to our clinic and consent for liver transplantation. All data were anonymized prior to analysis. Due to this and the retrospective design of the study, ethical approval was waived.

## 3. Results

### 3.1. Baseline Characteristics

Included were all pediatric liver transplant recipients in our center between 1 January 1983 and 1 October 2022 whose indication for transplantation was a malignant liver tumor. Altogether, our cohort consisted of 39 patients (making up to 4.2% of the total number of transplantations), 31 had hepatoblastoma, 8 other liver tumors, of which hemangioendothelioma and sarcoma were the most common (see Table 1).

Mean follow-up time was 10.2 years after diagnosis and 9.5 years after liver transplantation, with a significantly longer observation period for the patients with other liver tumors than hepatoblastoma. The majority of hepatoblastoma patients (65%) have been transplanted 2013 or later, whereas most of the other liver tumor patients (75%) were transplanted until 2012.

Of the 39 patients, 16 (41%) were female. Diagnosis of the tumor was made at a mean age of 3.4 years, while transplantation took place at an age of 4.1 years. Diagnosis and transplantation occurred significantly earlier in the group of hepatoblastoma patients (see Table 2).

### 3.2. Incidence of Pediatric Liver Transplantation for Malignoma over Time

Overall, 907 pediatric liver transplantations have taken place in our center over the last 40 years, with in 39 (4.3%) of the cases a primary hepatic tumor being the indication for transplantation. Incidence of pLTx for malignoma rose over time, with more than half of the liver transplantations for malignoma taking place in the current decade (2013–2022), whereas pediatric liver transplantations in our centre have not shown a further increase during the last 10 years. Overall, the share of liver malignoma as indication for pLTx compared to the pediatric liver transplantations in total has risen from 1.9% from 1983–1992 to 9.1% (2013–2022) (see Figure 1) and when corrected for expected incidence (hypothetical probability) according to total number of transplantations, have shown a significant difference (*p* = 0.0004). Especially, the share of high risk hepatoblastoma patients has been rising, with, for example, extrahepatic metastases being manifest in 62% of patients in the current cohort (2013–2022) and only 38% between 1983–2012.

### 3.3. Treatment before Liver Transplantation/Salvage Transplantation

Tumor extent exceeded the liver in more than half of the patients (51.3%), with distant metastases being evident in 10 patients (25.6%). Nineteen patients had an operation before liver transplant, with partial liver resection/hemi-hepatectomy in eight cases and the removal of pulmonary metastases in seven patients being the most common indications. Six transplantations were salvage transplantation patients who received a liver transplant for tumor recurrence after liver tumor removal, two patients had to be transplanted after liver surgery without tumor recurrence (one patient developed acute liver failure due to vascular complications, one patient developed chronic liver failure). Of the eight salvage transplantations, six were patients with hepatoblastoma, one patient had hepatocellular carcinoma and one further patient had angiosarcoma. Of the hepatoblastoma patients, three patients were staged PRETEXT II and three were PRETEXT III; three of the patients with hepatoblastoma had extrahepatic tumor manifestations. Histopathologic details on resection margins were available in five patients; all were R0 resections; in three cases tumor tissue was described as “close” to the resection margin (in one case only 1 mm). Histopathology revealed mesenchymal-epitheloid and mixed embryonal-fetal hepatoblastoma in five cases, and overly fetal hepatoblastoma in one case.

Of the hepatoblastoma patients categorized in the classification, fifteen had stage PRETEXT IV and five patients were in stage PRETEXT III at diagnosis. Eleven of these patients have been staged retrospectively as they were diagnosed before introduction of the SIOPEL PRETEXT classification.

Most of the patients (92.3%) received neo-adjuvant chemotherapy; the share of untreated patients was significantly lower in the group of hepatoblastoma patients. Study protocols used were HB’89, HB’93 and SIOPEL 3 and SIOPEL 4. The medical and surgical treatment of hepatoblastoma patients was coordinated with the German hepatoblastoma study center in Munich.

### 3.4. Perioperative Course/In-Patient Stay for Liver Transplantation

Patients were transplanted after an average overall waiting time on the EUROTRANSPLANT liver transplantation list of 41 days. The average waiting time in high-urgency cases was 9.7 days.

Patients with hepatoblastoma had a significantly lower length, weight and BMI at the time of transplantation when compared to the patients with other liver tumors, correlating to the younger age of these patients.

Laboratory values at the time of transplantation showed no signs of liver function impairment or cholestasis, except for the two patients being transplanted for liver failure after tumor resection. Renal function was normal.

One patient received living-related liver transplantation. Most of the patients received liver segment transplantation; only 40% of patients received a full-sized graft.

For the liver transplantation, mean in-patient stay in ICU was 9.7 days and non-ICU-stay was 26 days, with similar values for both hepatoblastoma and other liver tumor patients.

### 3.5. Outcome after Liver Transplantation

#### 3.5.1. Immunosuppression and Episodes of Rejection

After liver transplantation, nearly all patients (94.9%) received immunosuppression with Cyclosporine A and prednisolone, but the immunosuppressive regime changed in most of the patients (61.5%) with only one third of the patients still receiving Cyclosporine monotherapy as maintenance therapy. The most used maintenance immunosuppression was Sirolimus with 41% of patients, Tacrolimus was used by 15% of the patients. The main reasons for the change of immunosuppression were rejection episodes and renal function impairment.

Thirteen patients had one acute biopsy proven rejection (ABPR), two children had two episodes of rejection, two patients had three and another two patients even had four ABPR; so altogether more than half of our patients (51.3%) had at least one episode of ABPR.

Maintenance immunosuppression had no influence on the incidence of rejection episodes, with 53% of rejections occurring under m-Tor-inhibitors and 50% in the patients with CNI-based immunosuppression (*p* = 0.855).

#### 3.5.2. Biliary Complications

Nearly half of the total group of liver tumor patients had a post-transplant impairment of bile excretion requiring intervention, with a significantly higher share of patients in the group of hepatoblastoma patients (58.1 vs. 12.5%) and non-significant higher proportion of patients with bilio-digestive anastomosis compared to direct bile duct anastomosis (59.1% vs. 35.1%; *p* = 0.1382). Furthermore, biliary complications occurred more often in patients receiving a split transplantation (58.3 vs. 33.3%; *p* = 0.1258), but again without significant difference. Overall, the risk of biliary complications in the patients transplanted for liver tumor was significantly higher than in the 910 patients of our total transplant cohort (48.7% vs. 24%, OR 2.533; 1.874–3.424; *p* < 0.0001). Thirteen of the patients needed surgical revision of the biliary anastomosis, six required ERCP and stenting, two patients received PTCD and two patients required re-transplantation due to cholestatic graft fibrosis; with four of these patients requiring more than one intervention.

#### 3.5.3. Hearing Loss

Fifteen patients (38%) had a relevant hearing loss and received a hearing aid from our paedaudiologic specialists. All of these were patients with a hepatoblastoma, so nearly half of the patients with hepatoblastoma (48.4%) suffered from relevant hearing loss requiring hearing aids. All patients with hearing impairment had received neo-adjuvant chemotherapy before transplantation, and 14 of 15 patients received post-operative chemotherapy. Hearing impairment was diagnosed before liver transplantation in five patients, in ten patients it was diagnosed after transplantation (and adjuvant chemotherapy). The average number of Cisplatin cycles was 4.1 in patients without hearing loss and five cycles in patients that developed hearing loss, but without reaching a level of significance (*p* = 0.816). None of the patients received sodium thiosulfate.

#### 3.5.4. Secondary Malignoma and Tumor Recurrence

Nine patients (23.1%) developed a secondary malignoma during the observation period after liver transplantation; in six patients this occurred by recurrence of the primary tumor in the liver or metastases of the primary tumor. Of the three patients being diagnosed with a new tumor entity, one patient developed an ovarial teratoma, one patient skin cancer and one patient a multi-focal blastic tumor in the abdomen. We did not observe any cases of PTLD in our patients.

#### 3.5.5. Long-Term Outcome

Five-year-survival in our patients was 84.4% for 32 of the 39 patients, with 7 more patients who are currently alive but have not completed a 5-year follow-up yet. Survival was 81.1% in the hepatoblastoma group and 100% in the non-hepatoblastoma group. Six patients died after liver transplantation; all were patients diagnosed with hepatoblastoma. The cause of death was tumor recurrence in four patients, a de-novo malignancy in one patient and sepsis with multi-organ failure in one patient. Mean survival of the deceased patients was 2.4 years after diagnosis and 1.7 years after liver transplantation.

Four patients needed re-transplantation, because of cholestatic fibrosis of the graft in two cases, initial non-function of the graft in one case and arterial thrombosis of the graft in another.

### 3.6. Risk Factors for Tumor Recurrence and Mortality

In our nominal-logistic regression analysis, we analysed different factors for a possible influence on tumor recurrence and mortality (see Table 3). The only highly significant risk factor for mortality was tumor recurrence. A risk factor for tumor recurrence was liver transplantation as a salvage transplant. Further on, a lower decrease from the maximum AFP value to the AFP value measured before transplantation in relation to and a higher absolute AFP-value at the time of transplantation increased the risk of tumor recurrence. This analysis was only performed for the patients diagnosed with hepatoblastoma. Even though there was no statistically significant difference, AFP response measured as a quotient of AFP_max_ to AFP_pre-pLTx_ was higher in patients with primary transplantation (784) when compared to the patients with salvage transplantation (227; *p* = 0.263). Three of eight patients (37.5%) died after salvage transplantation compared to three of 31 patients (9.7%) after primary transplant, but the difference was not statistically significant. The diagnosis, time span from diagnosis to transplant, chemotherapy, rejections and immunosuppression did not have a significant influence on mortality or recurrence of the tumor.

## 4. Discussion

In our retrospective single center analysis, we have analysed pediatric liver transplant recipients who have been transplanted for malignant liver tumors over the last 40 years in the EUROTRANSPLANT region.

The main findings show a rising share of malignant diseases in the total number of pediatric liver transplants, a higher number of biliary complications compared to our total transplant cohort and an unexpectedly high incidence of rejections after transplantation. Factors with a negative impact on the long-term outcome are a reduced response to neoadjuvant chemotherapy as expressed by a smaller decrease of AFP in the subgroup of hepatoblastoma patients and transplantation as salvage transplantation. Furthermore, mTOR-inhibitor-based immunosuppression is of increasing importance and the most commonly used maintenance immunosuppression in our patient cohort.

The 5-year survival of 84% in our cohort is in line with the improving results of the current studies [10,11,16], but it has to be taken into account that our patient cohort dates back to 1983. However, in contrast to other studies, we could not observe a significant increase in 5-year survival in the current era when compared to previous transplantations [9]. This might be related to the fact that in our center, we find a concentration of complex patients in the current Era, often with metastatic disease (64% from 2013–2022 vs. 38% from 1983–2012) who might not have been eligible for transplantation in earlier eras.

Overall, the share of patients transplanted for malignoma in relation to the total pediatric liver transplant activity has more than quadrupled when compared to the first decade of our observation period (1983–1992); this trend can be observed from most large pediatric liver centers or multicenter studies for these eras [5,7,11]. One reason, certainly, is an overall increased incidence of hepatoblastoma and earlier referral for liver transplant evaluation [3,4]. Furthermore, this might be explained by a change of transplant listing criteria over the previous years as well as increased survival rates even from metastatic disease due to improvements in oncologic and surgical treatment; allowing many more children to achieve the chance of liver transplantation [18].

In our group, we found a higher overall survival rate in the patients with non-hepatoblastoma liver malignancy (100% vs. 80.6%); this stands in contrast to most other studies, but probably might be explained by the fact that, in our center, non-hepatoblastoma liver tumor patients were a small group consisting of three patients with hemangioendothelioma, three patients with sarcoma, one patient with inflammatory myofibroblastic tumor and only one patient with HCC, with the latter usually accounting for the main share of liver tumors, other than hepatoblastoma, and is associated with a higher mortality [7,11,18].

Hepatoblastoma patients were significantly younger at diagnosis and transplantation, which stands in line with most other studies.

We saw a higher share of patients with disease extension beyond the liver, especially in the current decade, which has increased the hazard ratio for mortality and tumor recurrence, but not to a significant level; supporting the current findings of our American colleagues [11,18].

Our patients with salvage transplantation had a significantly higher risk of tumor recurrence compared to the ones with primary liver transplant. There was also a higher mortality in this group (37.5% vs. 9.7%), but salvage transplantation did not reach significance as a risk factor for mortality even though tumor recurrence, which was more frequent in these patients, did. This underlines the findings of many previous studies, but some recent studies found similar results for both groups [11,15]. Arguably, with a partial hepatectomy, there might be a risk for macroscopically undetectable tumor cells that lead to recurrence of the tumor or metastases. Potentially, children who have already experienced a recurrence of a tumor might be suffering from a more aggressive subtype of hepatoblastoma, and might develop secondary resistance due to more cycles of chemotherapy and thus be at higher risk of another relapse. To support this, in a subgroup analysis, the patients with salvage transplantation showed a lower response (though not significant) in terms of AFP decrease before transplantation, which might argue for tumor biology as a driver of recurrence rather than surgical treatment. Data are contradictory and our subgroup of patients with salvage transplants covers more than 30 years of development in pediatric oncology and liver transplantation and is too small to draw definite conclusions.

In our study, we see a high share of bile flow impairment after liver transplantation requiring intervention (49%), which clearly exceeds the number of biliary complications in our total transplant cohort (24%) [6]. The described percentage of biliary complications in the SPLIT registry patients transplanted for liver tumors is also lower [11], even though it must be stated that in this study complications were only observed for one year whereas in our group, complications of the whole observation period were included. To our knowledge and research, there is no literature supporting the data. As a possible explanation, infiltrative growth of the tumor beyond the liver might complicate and impact surgical techniques for biliary anastomosis. Pre- and post-operative chemotherapy might lead to leukopenia and injury of biliary tissue, even though scarce literature in adult tumor patients could not show a significant difference in biliary complications between patients with neo-adjuvant chemotherapy and those without [19]. The rate of biliary complications in pediatric liver transplant recipients in general in the literature reaches comparable levels (26–41%) [8,20].

Furthermore, the rate of rejections in our group was rather high with more than half of the patients having at least one episode of acute biopsy-proven rejection. This also is in the high range when compared to the actuarial overall rates of ABPR after pediatric liver transplantation in general [8,21] and stands in contrast to the general assumption that due to the chemotherapy, oncologic patients have a lower risk of rejection episodes. It might be related to the fact that nearly all of our patients received Ciclosporine A as primary immunosuppressant and that in the historic cases, target CSA levels were lower than in other patient groups, in order to minimize immunosuppression and thus the risk of secondary malignant disease.

In the further course, more than half of the patients switched immunosuppression to other immunosuppressants, mainly Sirolimus, which showed a sufficient protection from rejection in most cases; and furthermore, indicated a tendency for a lower risk of tumor recurrence, even though it did not reach a significant level. Other studies have shown m-TOR-Inhibitors to be promising immunosuppressants after pediatric liver transplantation in general, especially in oncologic patients and to preserve renal function [22,23,24]. In our study cohort, Sirolimus seems to be a safe and effective immunosuppression, possibly with a positive effect on the risk of tumor recurrence at a similar rejection rate, though its more common adverse effects on bone marrow toxicity, wound healing impairment and aphthous mucosal lesions must be kept in mind [22,25].

We see a high number of patients with hearing loss, requiring hearing aids in our group, which is obviously a typical effect of the ototoxic chemotherapy regimen for hepatoblastoma [26,27]. The comparably high proportion of patients in our cohort suffering from cisplatin-induced hearing loss (CIHL) is associated to the rather high doses of chemotherapy in patients with metastatic disease and many patients receiving post-operative chemotherapy to prevent tumor recurrence. As hearing impairment is differently defined in different studies (hearing test abnormalities, requirement of hearing assistive devices), the total amount of patients is not easily compared and further data in this field are welcome. The treatment and prevention of CIHL is addressed in current studies [28,29] and hopefully will be at least partially prevented in the future with the use protective medication.

The only analysed factors with significant impact on tumor recurrence and mortality in our group of patients were the total AFP at the time of transplant and the relation of the highest measured AFP value before treatment and the AFP value after chemotherapy at the time of liver transplantation. AFP obviously is a good marker for chemotherapy response and thus reduction of tumor mass, and therefore a high value at the time of pLTx and a lower decrease during therapy could be correlated to the risk of tumor recurrence and thus survival. This has also been observed in the analysis of the SPLIT registry [11].

Our study has limitations—data was collected over a long time period of nearly 40 years, over which tumor classifications, oncologic and surgical therapies and liver transplantation for children in general have undergone substantial changes. Furthermore, our group of non-hepatoblastoma liver malignancies is quite heterogenous and therefore hard to compare to other studies. Furthermore, due to the retrospective design of the study, we could not gather data regarding tumor biology and histopathologic and genetic sub-analyses that influence course of disease, response to therapy and risk of recurrence. Our total group size—though not small for a group of liver transplantations for childhood liver malignancies—is relatively small, so some statistical tests with subgroups obviously have a limited validity.

In conclusion, liver transplantation for hepatic malignancy in children has very good 5-year survival rates, exceeding 80%. In our study, the patients who received salvage transplantation had a higher risk of tumor recurrence and with this a higher risk of mortality; however, with the comparably low patient numbers and the large time span, in addition to the substantial changes in oncologic and surgical therapy, these results must be interpreted with caution. The decision between primary transplant and primary partial hepatectomy as surgical therapy after chemotherapy must continuously be made very carefully under consideration of individual tumor extent and response to chemotherapy. Close AFP monitoring and especially the ratio of maximum AFP to post-chemotherapy AFP values can help to predict the risk of tumor recurrence after transplantation. The comparably high rate of biliary complications and rejections in our patient cohort needs to be confirmed in other cohorts, but the latter could raise the question of a more effective immunosuppression after transplantation, for which m-TOR-inhibitors might be a promising option.

## Figures and Tables

**Figure 1 children-10-00202-f001:**
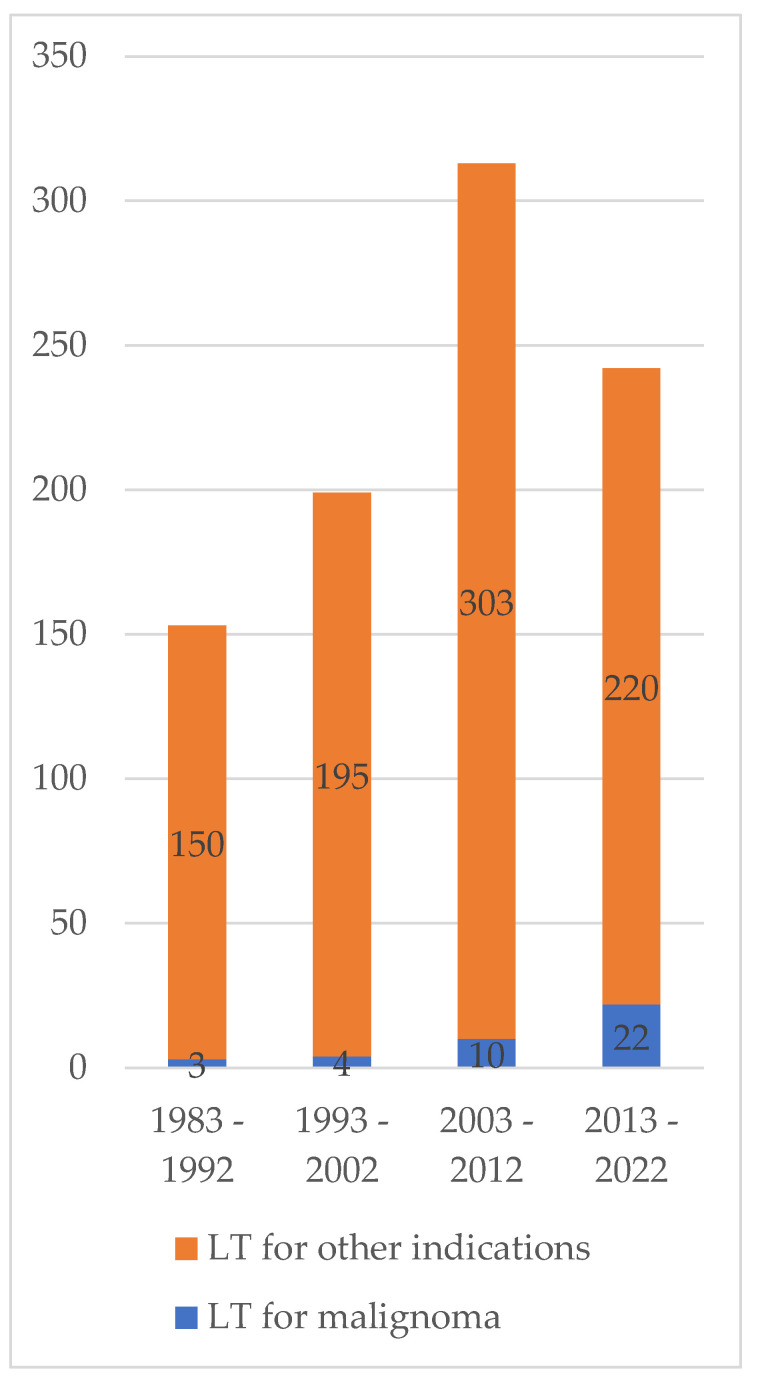
Incidence of liver malignancy as indication for liver transplantation in comparison to total liver transplant activity. Note that, in 2022 liver transplantations were only included until 1 October 2022.

**Table 1 children-10-00202-t001:** Patient characteristics, categorical variables. Divided into subgroups of hepatoblastoma and other primary liver malignancies as expressed using mean and standard deviation for whole group and sub-groups as well as median and range only for the whole group.

Categorical Variables	Total (n = 39)		Hepatoblastoma (n = 31)		Liver Tumor Other than Hepatoblastma (n = 8)		*p*-Value
	N	%					
Diagnosis							
Hepatoblastoma	31	79.5	31	100	0		
Hemangioendothelioma	3	7.7	0		3	37.5	
Sarcoma	3	7.7	0		3	37.5	
HCC	1	2.6	0		1	12.5	
Inflammatory myofibroblastic tumor	1	2.6	0		1	12.5	

Tumor extension surpassing liver	20	51.3	19	61.2	1	12.5	**0.0105**
Neoadjuvant chemotherapy	36	92.3	30	96.8	6	75	**0.0393**
Chemotherapy after liver transplant	26	66.7	23	74.1	3	37.5	**0.0342**
Salvage transplantation	8	20.5	6	19.4	2	25	0.7244

PRETEXT II	3	7.7	3	9.7			
PRETEXT III	10	25.6	10	32.3			
PRETEXT IV	18	46.2	18	58.1			
Sex female	16	41.0	12	38.7	4	50.0	0.5627
Era of Transplantation [1,2,3,4]							
Era 1 (1983–1992)	3	7.7	2	6.4	1	12.5	**0.0250**
Era 2 (1993–2002)	4	10.3	1	3.2	3	37.5	
Era 3 (2003–2012)	10	25.6	8	25.8	2	25	
Era 4 (2013–2022)	22	56.4	20	64.5	2	25	
On high-urgency list [y/n]	**33**	**84.6**	**28**	**90.3**	**5**	**62.5**	**0.0436**
Combined pLTx to other organ	1	2.6	1	3.2	0	0	n/a
Kidney	1	2.6	1	3.2	0	0	n/a
Full size graft	14	39.9	10	32.3	5	62.5	0.1170
Hepaticojejunostomy as biliary anastomosis	22	56.4	19	61.3	3	37.5	0.2263
Bile flow impairment requiring intervention	19	48.7	18	58.1	1	12.5	**0.0215**
Initial immunosuppression							n/a
CSA/Prednisolone	37	94.9	30	96.8	7	87.5	
Tacrolimus	1	2.6	0	0	1	12.5	
CSA/MMF	1	2.6	1	3.2	0	0	
Last documented immunosuppression							0.3124
CSA	13	33.3	10	32.3	3	37.5	
Sirolimus	16	41.0	14	45.2	2	25	
Tacrolimus	6	15.4	4	12.9	2	25	
MMF	2	5.1	2	6.4	0	0	
MMF + Predni	1	2.6	0	0	1	12.5	
CSA + Everolimus	1	2.6	1	3.2	0	0	
Acute biopsy-proven rejection episode	20	51.3	14	45.2	6	75	0.1322
Hearing loss	15	38.4	15	48.4	0	0	**0.0083**
Renal function impairment [GFR < 60]	6	15.4	4	16.1	2	25	0.5020
Secondary malignoma	9	23.1	8	25.9	1	12.5	0.4258
Graft loss due to death or re-pLTx	9	23.1	7	22.6	2	25	0.8849
Subsequent re-Tx	4	10.3	2	6.5	2	25	0.1231
Mortality after transplantation	6	15.3	6	19.4	0	0	0.0820

**Table 2 children-10-00202-t002:** Patient characteristics, continuous variables. Divided into subgroups of hepatoblastoma and other primary liver malignancies as expressed using mean and standard deviation for whole group and sub-groups as well as median and range only for the whole group.

Variable	pLTx for All Liver Tumours (n = 39)		pLTx for Hepatoblastoma (n = 31)		pLTx for Liver Tumours Other than Hepatoblastoma (n = 8)		*p*-Value
	Mean (Median)	SD (Range)	Mean	SE	Mean	SE	
Age at diagnosis	3.35 (1.69)	3.45 (0.28–12.12)	2.73	3.05	5.74	4.07	**0.0129**
Age at pLTx [years]	4.05 (2.59)	3.69 (0.56–14.34)	3.35	3.05	6.77	4.86	**0.0088**
Time between diagnosis and pLTx [years]	0.72 (0.47)	0.76 (0.05–4.00)	0.62	0.27	1.03	0.27	0.1912
Weight at pLTx [kg]	17.01 (12.15)	11.65 (7.3–60.0)	14.70	1.92	27.25	4.04	**0.0081**
Height at pLTx [cm]	96.4 (86.5)	25.7 (63–169)	92.6	4.44	113.0	9.35	**0.0282**
BMI [kg/m^2^]	16.75 (16.51)	2.17 (13.0–23.5)	16.25	0.34	19.01	0.72	**0.0014**
AFP maximum [IU/L]	n/a		541,945 (338,065)	717,927 (526–3,114,000)	n/a		
AFP at pLTx [IU/L]			48,458 (108)	251,033 (2–1,400,000)			
AFP max/AFP LTx			9270 (719.9)	17,034 (1.33–60,000)			
Creatinine at pLTx [µmol/L]	47.3 (27)	112.9 (10–692)	34.2	46.7	50.0	21.2	0.7594
Bilirubine at pLTx [µmol/L]	21 (5)	68 (3–390)	23.1	12.8	9.0	30.8	0.6757
Albumine at pLTx (g/L)	39.4 (40)	5.6 (25–48)	39.5	1.08	39.0	2.86	0.8742
INR at pLTx	1.12 (1.13)	0.14 (0.9–1.69)	1.12	0.03	1.10	0.07	0.8063
Waiting time for pLTx all status [days]	41.2 (37)	16.7 (2–127)	36.9	9.59	55.5	35.9	0.3613
Waiting time for pLTx on high urgency (HU) list (only HU patients) [days]	9.72 (9)	7.1 (0–23)	10.17	1.35	8.12	2.53	0.4795
ICU stay post pLtx [days]	9.07 (6)	12.3 (1–65)	9.20	2.79	8.71	4.72	0.9301
NON-ICU stay post Ltx [days]	26.44 (22)	16.23 (0–68)	25.70	3.69	28.57	6.24	0.6953
Follow-up after pLTx [years]	9.49 (7.97)	9.14 (0.84–36.76)	7.59	1.46	16.87	2.86	**0.0064**
Follow-up after diagnosis [years]	10.20 (9.49)	8.84 (0.28–34.08)	8.21	1.50	17.89	2.95	**0.0059**

**Table 3 children-10-00202-t003:** Hazard ratios for survival/tumor recurrence analysed using nominal logistic regression and censored by death of the patient/tumor recurrence.

Variable	HR for Mortality				HR for Tumor Recurrence	
	Odds Ratio	95% Confidence Interval	*p*-Value	OR	95%CI	*p*
Hepatoblastoma	3.024	0.842–16.352	0.0820	1.346	0.134–13.474	0.7961
Tumor extension beyond liver	5.667	0.593–54.114	0.1319	4.620	0.510–41.887	0.1736
Previous surgery before pLTx	6.786	0.711–64.723	0.0961	2.400	0.385–14.968	0.3356
Waiting time for pLTx	1.002	0.994–1.010	0.6118	0.999	0.989–1.009	0.8977
Age at transplantation	1.007	0.796–1.274	0.9536	1.039	0.831–1.299	0.7381
Time span between diagnosis and transplantation	1.723	0.699–4.242	0.2370	1.822	0.717–4.631	0.2072
Current Era (2013–2022)	0.736	0.129–4.210	0.7313	1.667	0.267–10.394	0.5844
**Salvage transplant**	4.283	0.864–21.241	0.0749	**6.116**	**1.020–36.683**	**0.0476**
Neo-adjuvant chemotherapy	0.483	0.038–6.111	0.5739	0.323	0.024–4.255	0.4179
Log (AFP max.)	1.229	0.475–3.182	0.6612	1.370	0.465–4.035	0.5482
**Log (AFP pre-pLTx)**	1.854	0.941–3.651	0.0587	**2.327**	**1.049–5.168**	**0.0378**
**Log (AFP max/AFP pre-pLTx)**	0.498	0.219–1.132	0.0729	**0.371**	**0.136–0.926**	**0.0254**
Full size graft	1.818	0.379–8.731	0.4552	1.750	0.304–10.075	0.5309
Biliodigestive anastomosis	1.666	0.267–10.394	0.5777	1.372	0.278–6.775	0.6975
Biliary complication requiring intervention	0.563	0.114–2.773	0.4796	0.471	0.076–2.932	0.4194
Post-operative chemotherapy	0.909	0.142–5.809	0.9198	0.900	0.183–4.429	0.8972
Maintenance immunosuppression CSA	4.600	0.721–29.332	0.1064	2.000	0.346–11.583	0.4392
Maintenance immunosuppression Sirolimus	0.240	0.025–2.286	0.2146	0.152	0.017–1.369	0.0953
Episodes of ABPR	0.938	0.198–4.437	0.9352	0.941	0.165–5.361	0.9456
Renal function impairment	0.628	0.062–6.329	0.6935	0.960	0.091–10.099	0.9729
**Recurrence of tumor**	**15.896**	**2.886–87.567**	**0.0015**	**n/a**		

## Data Availability

Data are available upon reasonable request.

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
