# Peer review of "Long-Term Outcome Following Liver Transplantation for Primary Hepatic Tumors—A Single Centre Observational Study over 40 Years"

_children, 2023, doi:10.3390/children10020202_

Round 1

Reviewer 1 Report

I thank the editors and authors for the opportunity to review this interesting and well-written mansucript. Studies on indications, outcomes and risk factors of pediatric liver transplantation for hepatic malignancy are highly needed. 

The 40-year experience of the Hannover group with oncologic pediatric liver transplantation ist presented. Possible improvements are the following:

1. Please check table 1 for the numbers concerning tumor entity. There are conflicting information in table and text (2 or 3 sarcomas?). Please fill in all tumor entities in table 1 for better overview. 

2. The question of salvage transplantation and its outcome is highly relevant for decision making on primary transplant or resection in hepatoblastoma. Salvage transplantation is generally defined as either transplantation after tumor recurrence or after failed resection due to surgical complications, posthepatectomy liver failure etc. Details on those patients with salvage transplant would be highly interesting. This would include PRETEXT, multifocality, resection margins, histology subtype etc. I feel this would significantly increase the impact of this part. 

3. The high rate of biliary complications is indeed surprising. Was there any difference in the rate between different graft types (full/split?). 

I congratulate the authors for this interesting manuscript and hope that my proposal for improvement find their appreciation.

Author Response

Thank you very much for the review, please see the attachment for my poin-by-point-response, you will find our point-by-point response in blue

Reviewer 2 Report

The aim of the study is to determine long term outcome in pediatric liver transplant recipients for liver tumors an identify risk factors for mortality and tumor recurrence. The study looks at a long period of time in order to get sufficient numbers to try and analyze. The findings themselves were not very novel, but do add to the existing literature. The English could be improved upon in some areas - some sentences are confusing to read. 

Major comments:

1) The title - doesn't quite make sense to talk about biliary complications and hearing loss when the project is about a lot more than that. Perhaps talking about outcomes after liver transplantation for primary hepatic malignancy rather than those two specific things

2) Line 94-95 - Can you explain why patients with chronic liver disease who develop liver tumor (more likely HCC) would bias results? How many of those patients were there? Would be helpful to explain why that would bias the results (i.e. there is data to support that they have better outcomes than those transplanted for liver tumors outside of a pre-existing condition, etc).

3) Definitions - would include PRETEXT and POSTTEXT within definitions with references. 

4) Line 157 - 158 - You comment on the percent of transplants for malignancy compared to total transplants, but don't actually give the numerical number of total transplants in the results - it can be inferred from the graph, but would be nice to have it written out as well. 

5) Table 1 - liver tumor other than hepatoblastoma - for the diagnoses, you have n=8 overall but only 6 are in the table and as such, the % do not add up to 100%. Please review and make necessary changes. 

6) Table 1 - When you do statistical analyses for era of transplantation between hepatoblastoma and other liver tumors, you compare the groups as a whole - is this through ANOVA? If so, that wasn't detailed in the statistical plan.

7) Just a point - Statistical difference in hearing loss between the two groups may also be impacted by excluding those with pre-existing conditions to liver cancer (typically HCC) that may have also received cisplatin based therapy. 

8) line 181-182 - 11 patients didn't have PRETEXT scoring because it was before the classification existed, but could you not go back and PRETEXT those patients? 

9) line 229 - The hearing loss part doesn't really fit with the story. First off, it's entitled hearing loss after liver transplantation, implying that hearing loss is related to transplant, which it's not. I don't think the hearing loss should be included in the manuscript unless you are willing to delve a lot deeper into it. We know that cisplatin causes hearing loss and that it's often related to cumulative dose. So what was the cumulative dose of cisplatin? Was sodium thiosulfate used in any patients? Was hearing loss evident before transplant (with neoadjuvant chemotherapy) or after (adjuvant)? 

10) line 246-247 - Don't understand why the renal function is included in the manuscript - also doesn't fit with the paper, in my opinion, unless you are willing to delve further. In the patients with renal dysfunction, what was their cumulative dose of cisplatin? Did they develop renal dysfunction before or after transplant? If after transplant, was it perhaps because of immunosuppression and supratherapeutic cyclosporine levels? Did renal dysfunction effect outcome in terms of mortality or tumor recurrence? 

11) line 280-283, line 294-295, line 307-310- none of this data was included in the results section. No new information should be presented in the discussion. 

12) Paragraph on salvage transplant (starting line 307) - in the patients experienced recurrence after salvage transplant vs primary liver transplant - was there a difference in their AFP response? Or did they all have similarly poor AFP response? Depending on the answer to this, it may be biology rather than local control method that is driving their risk for recurrence. 

12) Another limitation of the study is the lack of pathologic correlation to outcome within the hepatoblastoma cohort. We know that biology of tumors is often the driving force in chemoresponsiveness and outcome. So when we talk about how patients who required salvage transplant had statistically higher risk of tumor recurrence, it probably speaks to the biology of those tumors (which you comment on in the discussion, but I think highlighting that limitation is important). 

Minor comments:

1) line 62 - I think tumor "extent" makes more sense than tumor extension

2) line 65 - would reword "worse results of the transplantation" to either "worse results with salvage transplantation" OR "worse results than upfront transplantation."

3) line 153 - pTLx never defined in the text

4) Table 2 - "pLTX for liver tumors other than hepatoblastoma" needs (n=8) added to it 

5) Table 2 - what does HU mean in "Waiting time for pLTx on high urgency list (only HU patients)

6) Line 173 - when you say "tumor extension exceeded the liver" do you mean contiguous tumor extension beyond the liver capsule or non-contiguous metastatic disease (i.e abdominal lymph nodes or more distant lung mets).  

7) line 194 - would change "according" to "correlating"

8) line 237 -239 - sentence doesn't make sense, would reword. Also, please just give a % of patients that developed a secondary malignancy rather than saying "nearly every fourth patient."

9) Line 250-253 - I would reorganize this paragraph and put it after the first paragraph in long term outcomes when you talk about 5 year survival in your cohort. 

10) line 259 - you did not define pre-tx-AFP value - is this pre treatment AFP value or pre-transplant?

11) Line 259-260 - sentence doesn't make sense and should be reworded.

12) line 274-275 - saying that a reduced response to neoadjuvant chemotherapy based on lower drop in AFP was a negative impact on outcome is specific to the hepatoblastoma group (not the entire liver cancer population). As such, you need to add that disclaimer to the sentence. 

13) line 345 - Sirolimus is still considered a form of immunosuppression. Would consider re-wording this. 

Author Response

Thank you for the helpful review; please see the attachment for my point-by-point response, you will find my answers printed in blue

Round 2

Reviewer 2 Report

The authors have addressed all my previous comments and I do feel the paper looks much stronger and complete now. I have no further suggestions.